# Antibiotic Therapy in the Critically Ill with Acute Renal Failure and Renal Replacement Therapy: A Narrative Review

**DOI:** 10.3390/antibiotics11121769

**Published:** 2022-12-07

**Authors:** Alberto Corona, Dario Cattaneo, Nicola Latronico

**Affiliations:** 1Accident & Emergency and Anaesthesia and Intensive Care Medicine Department, Esine and Edolo Hospitals, ASST Valcamonica, 25040 Brescia, Italy; 2Unit of Clinical Pharmacology, ASST Fatebenefratelli Sacco University Hospital, 20157 Milan, Italy; 3University Department of Medical and Surgical Specialties, Radiological Sciences and Public Health, University of Brescia, 25100 Brescia, Italy

**Keywords:** antibiotics, pharmacodynamics, pharmacokinetics, renal failure, renal replacement therapy

## Abstract

The outcome for critically ill patients is burdened by a double mortality rate and a longer hospital stay in the case of sepsis or septic shock. The adequate use of antibiotics may impact on the outcome since they may affect the pharmacokinetics (Pk) and pharmacodynamics (Pd) of antibiotics in such patients. Acute renal failure (ARF) occurs in about 50% of septic patients, and the consequent need for continuous renal replacement therapy (CRRT) makes the renal elimination rate of most antibiotics highly variable. Antibiotics doses should be reduced in patients experiencing ARF, in accordance with the glomerular filtration rate (GFR), whereas posology should be increased in the case of CRRT. Since different settings of CRRT may be used, identifying a standard dosage of antibiotics is very difficult, because there is a risk of both oversimplification and failing the therapeutic efficacy. Indeed, it has been seen that, in over 25% of cases, the antibiotic therapy does not reach the necessary concentration target mainly due to lack of the proper minimal inhibitory concentration (MIC) achievement. The aim of this narrative review is to clarify whether shared algorithms exist, allowing them to inform the daily practice in the proper antibiotics posology for critically ill patients undergoing CRRT.

## 1. Introduction

The critically ill outcome is burdened by a double mortality rate and a longer hospital stay as the result of infectious complications. Approximately 51% of intensive care unit (ICU) patients are classified as infected, and 70% receive antibiotics [1,2] with an infection rate increasing from 30% to 70% between the first and seventh day of ICU stay. Critically ill patients experiencing infection appear to have more comorbidities and clinical conditions of greater severity upon ICU admission than non-infected patients, together with higher sequential organ failure assessment (SOFA) and simplified acute physiology (SAPS II) scores [1,2]. Renal replacement therapy (RRT) is one of the main risk factors for infection, together with mechanical ventilation, diabetes, chronic obstructive pulmonary disease, neoplasms, liver cirrhosis, and any type of immunosuppression. The main recorded sites of infection are the respiratory system (64%), the abdomen (20%), the circulatory system (15%), and the genitourinary tract (14%). Gram-negatives constitutes the more frequent isolates (62% compared to 50% of previous studies), against 47% of Gram-positive [1,2].

ARF is defined as a clinical condition that is usually associated with reduced daily urine output and overload of the extracellular compartment from fluid retention, and it is characterized by (i) a rapid decline in GFR occurring over a limited amount of time; and (ii) increased serum creatinine levels, accumulation of nitrogenous waste, and electrolyte and acid–base imbalances [3]. It is necessary to distinguish between simple kidney damage (injury), with only a minor malfunction, from an absolute impairment of organ function (failure). Acute kidney injury (AKI) developed during hospitalization and due to non-renal problems has a 5 to 10 times higher incidence than that acquired in the community [4] and occurs in about 36% of patients [5]. Acute tubular necrosis (ATN) is the most common (about 75%) cause of ARF in the ICU, followed by pre-renal forms of hypoperfusion (18%) and chronic exacerbation (8%) [6,7]. ARF develops in over 50% of critically ills with an associated mortality of 30% [8].

Optimal antibiotic therapy is essential in these patients, but the drug elimination rate during CRRT can be highly variable, depending on the method used, the characteristics of the same, and the patient’s condition.

Indeed, sepsis increases the drug volume of distribution (Vd), prolongs the half-life (t_1/2_), and alters the protein binding of many antibiotics, with increased capillary permeability due to the release of inflammatory mediators with fluid accumulation and hypoalbuminemia [9,10].

Antibiotics should be reduced in patients experiencing ARF, according to the glomerular filtration rate (GFR), whereas posology should be increased in the case of CRRT. Since different settings of CRRT may be used, identifying a standard dosage of antibiotics is very difficult, because there is a risk of both oversimplification and failing the therapeutic efficacy. Indeed, it has been seen that, in over 25% of cases, the antibiotic therapy does not reach the necessary concentration target mainly due to lack of the proper minimal inhibitory concentration (MIC) achievement. The aim of this narrative review is to clarify whether shared algorithms exist, allowing to inform the daily practice in the proper antibiotics posology for critically ill patients undergoing CRRT.

## 2. Results

### 2.1. Antibiotics in the Critically Ill

Antibiotics are divided into (i) bacteriostatic, which is capable of inhibiting bacterial growth and ensures the definitive elimination of the microorganism only with the contribution of the body’s immune system; and (ii) bactericidal action, which guarantees a bacterial survival equal to or less than 0.01% 24 h of “in vitro” contact (Table 1). Only antibiotics that act on fundamental structures for the bacterial cell, such as the wall or nucleic acids, will be bactericidal [11,12,13].

To assess whether an antibiotic is bacteriostatic or bactericidal, the following are determined [6]: MIC—minimum inhibitory concentration, the minimum concentration of antibiotic capable of preventing the development of microorganisms (µg/mL).MBC—minimum bactericidal concentration, the minimum concentration of antibiotic capable of leading to death of bacterial cells (µg/mL).

If the antibiotic is bactericidal, the MIC and MBC values coincide. If the antibiotic is bacteriostatic, the MIC and MBC values are different (MBC > MIC).

Antibiotics can be classified on the basis of the action site; we can identify [14] antibiotics inhibiting (i) cell wall synthesis, (ii) protein synthesis, and (iii) the replication and transcription mechanism of nucleic acid. Moreover, there are antibiotics that alter the bacterial or fungal cytoplasmic membrane as well as antibiotics that act as antimetabolites.

Antibiotics are also grouped into families with similar characteristics [14].

Beta-lactams are antibiotics that inhibit the synthesis of the bacterial cell wall, carrying out a bactericidal action. In turn, they are divided into subclasses.

Penicillins can be divided into drugs with or without beta-lactamase inhibitors, which improve their effectiveness in the case of microorganisms that have developed this mechanism of resistance. They work by inhibiting the synthesis of peptidoglycan. They are active against Gram-positive and some Gram-negative, in particular piperacillin–tazobactam, which also acts on *Pseudomonas* spp. They have poor intracellular penetration into the eye, prostate, and meninges in the absence of inflammation. Intracellular bacteria are inherently resistant and lack a bacterial wall. They include Ampicillin, Amoxicillin, Oxacillin, and Piperacillin.

Cephalosporins can be divided into generations (I–VI), with different activity on Gram-positive, Gram-negative, or both (higher spectrum). They act through irreversible inhibition of the transpeptidases involved in the synthesis of peptidoglycan. They spread well in most organs and tissues. They include ceftaroline, ceftriaxone, ceftazidime, cefepime, ceftolozane (plus tazobactam), and ceftaroline fosamil.

Cabapenemics and Monobactams are broad spectrum, also action on anaerobes, and have good tissue penetration. The main mechanism of resistance is linked to the production of carbapenemase by the bacterial cell. They include Imipenem, Ertapenem, and Meropenem. Monobactams include aztreonam.

Fluoroquinolones are bactericidal antibiotics that work by inhibiting the synthesis of bacterial DNA. They have a broad spectrum with action also towards intracellular cells. They include Ciprofloxacin, Levofloxacin, Moxifloxacin, and Dalafloxacin.

Aminoglycosides are bactericidal antibiotics that inhibit bacterial protein synthesis by binding to the 30S ribosomal subunit. They act mainly on Gram-negatives, with intrinsic resistance on anaerobes and *Enterococcus* spp. They have poor penetration of the blood–brain barrier, lung, and biliary tract. Gentaomicin and Amikacin are part of it.

Glycopeptides are bactericidal antibiotics that inhibit the synthesis of the bacterial wall with action on the peptidoglycan. The spectrum is restricted to Gram-positive, both aerobic and anaerobic. They have poor penetration into the lung parenchyma and bile. They are used in Gram-positive oxacillin-resistant infections. They include Vancomycin and Teicoplanin.

Lipoglycopeptides inhibit bacterial cell wall synthesis and disrupt cell membrane integrity; they include dalbavancin, oritavancin, and telavancin.

Lipopeptides are bactericidal antibiotics that are used for the inhibition of protein synthesis and nucleic acids through potassium efflux, cellular depolarization, with restricted action on Gram-positives. Daptomycin is part of it.

Rifamycins are bactericidal antibiotics used for the inhibition of transcription from DNA to mRNA, with action on DNA-dependent RNA polymerase. Broad spectrum, including atypical, with wide tissue distribution. Rifampicin is part of it, always in combination therapy due to the risk of developing resistance.

Nitroimidazoles possess bactericidal action for structural modifications of bacterial DNA, restricted to anaerobes and protozoa. Good tissue diffusion. Metronidazole is part of it.

Streptogramine includes Quinupristin and Dalfopristin, which inhibit bacterial ribosomal protein synthesis with action on Gram-positive and atypical.

Nitrofurans include nitrofurantoin, a bactericidal antibiotic for inhibiting the synthesis of nucleic acids, used mainly for urinary tract infections.

Macrolides are bacteriostatic but bactericidal antibiotics in high doses that inhibit protein synthesis by action on the 50S ribosomal subunit. Action on Gram-positive, Gram-negative, and intracellular, with poor penetration into the bone, urinary tract, and blood–brain barrier. They include Erythromycin, Azithromycin, and Clarithromycin.

Tetracyclines are bacteriostatic antibiotics for action on the 30S ribosomal subunit with inhibition of protein synthesis. Action on Gram-positive, Gram-negative, and intracellular. They include Doxycycline.

Glycylcycline exhibits bacteriostatic action by inhibiting protein synthesis via the 30S ribosomal subunit. Action on Gram-positive, Gram-negative, and intracellular. Good diffusion in most tissues but not in the blood–brain barrier. They include Tigecyclin.

Oxazolidinones include Linezolid, a bacteriostatic but bactericidal antibiotic if used in IC that works by inhibiting protein synthesis with action on the 50S ribosomal subunit. It also exhibits activity against some bacterial toxins, such as PVL (panton–valentine leukocidin). Action on Gram-positive.

Sulfonamides are bacteriostatic and used for the inhibition of the synthesis of tetrahydrofolic acid. Wide spectrum of Gram-positive and Gram-negative. The Trimethoprim–Sulfamethoxazole combination has bactericidal efficacy.

Lincosamides include Clindamycin, a bacteriostatic for the inhibition of the 50S ribosomal subunit and active on some strepto and staphylococcal bacterial toxins. Action on Gram-positive, anaerobes and protozoa.

Finally, antibiotics can be divided, based on their solubility, into hydrophilic and lipophilic. The former, which include beta-lactams, glycopeptides, and aminoglycosides, have a limited distribution to plasma and extracellular fluids, and they are usually excreted by the kidney. The latter, which include macrolides, fluoroquinolones, tetracyclines, linezolid, chloramphenicol, and rifampin, have a wide distribution in the intracellular compartment, with a greater distribution in the body.

### 2.2. Main Agents That Gained Market Authorization between 2017 and 2020 [15,16]

As of 2017, eight new antibiotics have been approved by the Food and Drug Administration [15]. 

Most of the approved compounds target carbapenem-resistant Enterobacteriaceae (CRE). However, both omadacycline and eravacycline are derivatives of tetracyclines. Omadacycline is a semisynthetic drug and has activities against Gram-positives, including difficult-to-eradicate MRSA and some Gram-negatives; it is approved for the treatment of community-acquired pneumonia (CAP). Eravacycline is totally synthetic and approved for the treatment of complicated intra-abdominal infections (IAI). New combinations of β-lactam antibiotic and β-lactamase inhibitor, such as ceftazidime–avibactam and meropenem–vaborbactam, presenting activities against *Klebsiella pneumonia* carbapenemase producing (KPC), have been approved since 2018. They attack peptidoglycan biosynthesis, interrupting the formation of the bacterial cell wall through covalent binding to penicillin-binding proteins (PBPs). However, new treatment options for carbapenem-resistant *Acinetobacter baumannii* (CRAB) and carbapenem-resistant *Pseudomonas aeruginosa* (CRPA) are still lacking, particularly for those producing metallo-proteinases (class B β-lactamases), albeit aztreonam–avibactam has been approved [15]. Imipenem–relebactam: relebactam is an active β-lactamase inhibitor against class A (including KPC) and class C β-lactamases. 

The last approved fluoroquinolonic, delafloxacin, has greater in vitro activity against many Gram-positive pathogens, including quinolone-resistant strains, as well as other fluoroquinolones, such as Finafloxacin and Zabofloxacin [15].

New active antibiotics against resistant pathogens that cause acute bacterial skin and skin structure infections (ABSSSIs) have been studied, especially for infections caused by MRSA. The most recently approved antibiotics include dalbavancin, tedizolid, oritavancin, and delafloxacin [15]. The new aminoglycoside in the treatment of infection caused by CRE is Plazomicin, which is a bactericidal antibiotic that works on Gram-negative aerobes and some Gram-positive and *Mycobacteria* spp. [15].

Cephalosporins act by blocking the synthesis of the bacterial wall. There are five generations of cephalosporins, each characterized by a precise antimicrobial spectrum that becomes wider and wider, reaching the fifth generation, also active on MRSA. The compounds belonging to the latter generation (ceftobiprole, ceftarolin, and ceftolozane) have been developed to specifically combat multi-drug-resistant (MDR) bacterial strains. Ceftobiprole, used to treat community-acquired pneumonia, is effective against methycillin-resistant *Staphylococci.* Ceftolozane, combined with the β-lactamase inhibitor tazobactam, is highly dedicated to CRE and *Pseudomonas aeruginosa*. Cefiderocol is part of the siderophore cephalosporins, a new class of drugs, of which this antibiotic was the first to be approved, i.e., by the FDA in 2019 and by EMA in April 2020, for urinary tract infection (UTI) caused by Gram-negative, CAP, and ventilator-associated bacterial pneumonia (VABP) [15].

At the end of 2020, there were 43 antibiotics in clinical development, of which, 15 were Phase I, 13 were Phase II, and 13 were Phase III [15].

### 2.3. Pk/Pd of Antibiotics in Critically Ill Patients

The optimal use of antibiotics is essential for the frequency and impact that the infectious disease has on the outcome in the critically ill patients, also given the increase in microbial resistance [13,15,16,17]. However, it has been found that 30% to 60% of prescriptions are inappropriate or suboptimal [11,13,14,15,16,17,18,19,20]. There is a direct correlation between mis-prescribing and over-prescribing antibiotics and the emergence of resistant microorganisms. Hence, there is a need to optimize antibiotic therapy, ensuring the best outcome for the patient and limiting the risk of selecting MDR. This involves the rapid identification of patients with infection; the use of reasoned empirical therapy; evaluating the possible pathogens involved with respect to the site of infection, comorbidities, and local situations, as well as the Pk and Pd characteristics of the antibiotic; and descaling as soon as possible, once the germ in question has been identified, with short-term therapies where possible [13]. For antibiotic therapy to be effective, it is important that the pathogen is sensitive to the drug and that the latter reaches the infectious site with an adequate concentration for a sufficient period of time [21].

Pk describes the absorption, distribution, metabolism, and elimination of the drug.

Important Pk parameters are as follows: Bioavailability, the amount of drug absorbed into the systemic circulation after administration (100% for the intravenous route).Volume of distribution (Vd) is a virtual volume in which the total amount of a drug present in the body should be uniformly distributed to obtain the same concentration measured in the plasma [22,23]. Hydrophilic drugs will have a reduced Vd, limited to the bloodstream, while lipophilic drugs will tend to accumulate in the body for penetration into cells and adipose tissue, with a high Vd.

It is an indispensable parameter for establishing the initial dose (D) of administration of a drug, given by the product of Vd for the desired plasma concentrations [C] and for body weight: D = C × Vd × body weight.

Moreover, other important parameters are as follows [22,23]:Drug clearance, the volume of blood purified in a given time interval.T_1/2_, the half-life of the concentrations reached, which is important for defining the interval between administrations. It is closely linked to clearance (Cl) and Vd: T_1/2_ = 0.693 × Vd/Cl.Protein binding, the portion of the drug linked mainly to albumin, with the free portion being the one that carries out the pharmacological activity.The area under the concentration–time curve (AUC), which reflects the exposure of tissues to the drug over time. The AUC is a Pk parameter given by the integral of a concentration–time curve or the sum of the trapezoids that can be drawn under the curve (trapezoidal method). This parameter is essential for describing the effect of drugs, as it reflects the exposure of tissues to the drug over time. The AUC (zero to infinity) represents total drug exposure as a function of time. Assuming a linear Pd with elimination rate, K, it can be shown that the AUC is proportional to the total amount of drug absorbed by the body. The constant of proportionality is 1/K. The AUC, if the elimination follows first-order kinetics, can be used to calculate the clearance according to the formula Cl = D/AUC [22,23].

In the critically ill patient, there are numerous modifications that can alter the Pk [24] of drugs and, in particular, of antibiotics [25]. In the critically ill patients the only reliable way of drug administration is the intravenous one, since the likely presence of edema or peripheral hypoperfusion makes unpredictable the effect of the drug given by subcutaneous/intramuscular way. Moreover changes in intestinal absorption due to alterations in gastric pH, peristalsis, perfusion or due to exclusion of portions of the intestine for blind loops or ostomies. Furthermore, the V_d_ is increased in the majority of critically ill patients due to dyscrasia and fluid overload. Consequently, for a given drug dose, the C_max_ will be lower on the one hand, but the T_1/2_ will increase, with a possible tendency toward accumulation. The amount of drug bound to proteins also changes, because on the one hand, there is a tendency toward hypo-albuminemia, with less binding especially for acid drugs; and on the other hand, there is an increase in alpha_1_-glycoproteins related to inflammation, with greater binding to basic drugs [13,26].

Last but not least, due to the presence of organ dysfunction, with the possible alteration of liver and kidney function, there is a reduction in clearance. All this affects the Pd of the antibiotic, or rather the relationship between its mechanism of action and the concentration achieved. Time and concentration-dependent drugs can be identified [13,26]. For the former, the effectiveness of bacterial elimination is linked to the time during which their concentration in circulation is maintained above the minimum concentration that in vitro prevents bacterial replication (MIC) (T > MIC, with a concentration of 4–5 times the MIC). For this type, continuous infusion or prolonged infusion at each administration is indicated, with at least 50% of the time above the MIC with an ideal of 100%. For the latter, on the other hand, it is linked to the peak concentration reached with respect to the MIC, with a target of 8–10 times (C_max_ > MIC or AUC/MIC). A single administration is therefore more useful. Some antibiotics also exhibit a post-antibiotic effect (PAE), i.e., the ability to suppress bacterial growth even when the concentration falls below the MIC. This is especially true for antibiotics that inhibit bacterial protein or nucleic acid synthesis. This effect is very present among concentration-dependent antibiotics, and it is less represented among time-dependent and among these more pronounced for carbapenems. The target for time-dependent antibiotics with a reduced PAE will be a T > MIC as prolonged as possible; for time-dependent antibiotics with a significant PAE, it will be maximizing the AUC [26] (see Table 2 and Table 3).

This shows how important it is to optimize antibiotic therapy according to Pk/Pd parameters, because this can lead to fewer therapeutic failures, reduced mortality, and the onset of resistant microorganisms [24].

## 3. Acute Renal Failure

### 3.1. Definition

As previously mentioned, ARF is defined as a clinical condition characterized by (i) a rapid decline in GFR occurring over a limited amount of time; (ii) increased serum creatinine levels, accumulation of nitrogenous waste, and electrolyte and acid–base imbalances, usually associated with reduced daily urine output, with overload of the extracellular compartment from fluid retention [27]. It is necessary to distinguish between simple kidney damage (Injury), with only a minor malfunction, from an absolute impairment of organ function (Failure).

The first attempt at classification took place in 2002, by the Acute Dialysis Quality Initiative (ADQI) [28], with the so-called RIFLE criteria, acronym of: (i) Risk of renal dysfunction, (ii) Injury to the kidney, (iii) Failure of kidney function, (iv) Loss of kidney function (v) End-stage renal disease. The two main and already known parameters of serum creatinine and urinary output are taken into consideration, attributing the same importance to both markers, considering them separately, in evaluating not the absolute values of creatinine, given their extreme inter-individual variability, but the variations from a baseline, thus allowing the inclusion of acute-on-chronic cases and the subdivision into five stages with increasing severity, underlining the dynamic vision of the pathology and its possible evolution [29,30].

To try to further improve the definition, identification and classification of acute renal failure, an independent cooperation network was formed in 2007, composed of experts from both the nephrological and interventional fields, the Acute Kidney Injury Network (AKIN), which has drawn up a new evaluation system for this pathology [30]. Also in this case we are based on the increase of creatininemia in relation to the baseline and on the reduction of the hourly urine production, but the stages are reduced to three, indicated by numbers and not by letters, the time criterion is introduced, because the identification of changes must occur within 48 h or less and, in order to further increase sensitivity, minimal increases in creatinine are considered sufficient (0.3mg/dl is sufficient to return to the first level) [30,31,32].

In 2012, the Kidney Disease Improving Global Outcomes (KDIGO) proposed a new definition of AKI, within the KDIGO guidelines [33]. Any one of the following conditions is sufficient:Increased (serum creatinine) sCr of ≥0.3 mg/dL (≥26.5 μmol/L) within 48 h;Increase in sCr to a value ≥1.5 times the baseline value determined at least within 7 days previously;Urinary volume <0.5 mL/kg/hour for at least 6 h.

If the baseline value is unknown, a 75 mL/min/1.73 m^2^ GFR should be attributed to the patient in the absence of CKD. As regards staging, KDIGO identifies three stages.

The main criticism of this classification system concerns the lack of a grading of the severity of the clinical condition and the use, once again, of extremely variable and influential parameters, such as serum creatinine and urinary output [33,34,35].

### 3.2. Epidemiology, Etiology, and Pathogenesis

Two forms of AKI must be distinguished: that acquired in the community developing outside the hospital and that acquired during hospitalization due to non-renal problems, which has a 5-to-10-times-higher incidence [4]. The first type is most often represented by an isolated organ deficit, while the second is frequently part of a more complex picture, with associated and multiple organ deficits. In the ICU, the most common type of finding is acute tubular necrosis (about 75%), followed by pre-renal forms of hypoperfusion (18%) and chronic exacerbations (8%) [6,7]. In the ICU, the most probable estimate of the occurrence of AKI still seems to be around 36% [5].

The causes of acute renal failure are classically divided into three main categories:Prerenal, which includes pathological conditions that result in hypoperfusion of the kidney, with reduced organ function in the absence of frank parenchymal damage;Renal or intrinsic, in which the organ is primarily affected by alterations;Postrenal, in which the kidney is subjected to the consequences of an obstruction of the urinary tract [5].

Acute renal failure due to prerenal causes can complicate any condition involving hypovolemia, low cardiac output, systemic vasodilation, or selective intrarenal vasoconstriction [8].

In some at-risk subjects, however, only slight decreases in perfusion may be sufficient, because the compensation mechanisms are already compromised. This is the case, for example, for elderly patients or those with pathologies that involve the loss of integrity of the afferent arterioles, such as in the course of hypertensive nephrosclerosis, diabetic vasculopathy, and marked atherosclerosis; or in case of use of drugs that interfere with the adaptation mechanisms, such as inhibitors of prostaglandin biosynthesis, e.g., non-steroidal anti-inflammatory drugs (NSAIDs), or the activity of the angiotensin converting enzyme (ACE), the ACE inhibitors, or angiotensin II receptor antagonists, sartans.

NSAIDs do not compromise GFR in healthy individuals but can trigger acute renal failure in patients with volume depletion or chronic renal failure, because, in this case, the GFR is maintained by the hyperfiltration of residual nephrons by the action of prostaglandins.

ACE inhibitors and sartans are dangerous in those conditions in which glomerular perfusion and filtration are particularly dependent on angiotensin II activity, such as in the case of bilateral renal artery stenosis, or unilateral if you are facing a kidney single or double function.

The prerenal form of acute renal failure is considered reversible, precisely because the organ parenchyma is undamaged. However, in the event of a major collapse of renal blood flow, it is possible that ischemic damage occurs up to the onset of acute tubular necrosis, which is the main form of intrinsic renal insufficiency. There is, therefore, a continuum between the first two categories of deficit

AKI can be the consequence of ischemic or toxic tubular damage, tubulointerstitial diseases, pathologies of the renal microcirculation and glomeruli or finally involving the renal vessels of larger caliber. Ischemic tubular damage shares part of the etiologies of the prerenal form, only tending to be of greater severity. To these are added the pathologies of the large renal vessels, with obstruction from atheromatous plaque, thrombosis, dissection of aneurysm, compression from mass, or vasculitis.

The tubules and interstitium can be affected by allergic diseases, for example, in response to βlactam, quinolone, and NSAID drugs; of infectious origin, such as in pyelonephritis; infiltrative type, in the case of lymphoma, amyloidosis, or sarcoidosis; inflammatory type, such as Sjogren’s syndrome or tubulointerstitial nephritis with uveitis; or obstructive type, which can be divided into an exogenous form, for drugs such as aciclovir and ganciclovir or metrotrexate, and an endogenous one, from myeloma proteins or uric acid.

Although many ICU patients with AKI of ischemic or nephrotoxic origin do not present morphological evidence of cell necrosis, it is commonly referred to as acute tubular necrosis (ATN) and associated with these two conditions, which represent the most frequent cause.

The course is divided into four phases [36], often preceded by a period of prerenal azotemia: initial, extension, maintenance, and recovery.

In the initial phase, lasting from hours to days, the GFR decreases due to the collapse of the glomerular ultrafiltration pressure, to which is added the obstruction of the tubules due to cellular flaking and necrotic debris, and the retrograde passage of filtrate. This is the consequence of the establishment of an adenosine triphosphate (ATP) deficiency, which triggers necrotic degeneration, particularly in those tracts of the tubule more sensitive to its depletion, such as the S3 segment of the proximal or the ascending tract of the loop of Henle for the high rates of active transport and for the localization in the renal medulla, in which the partial pressure of oxygen is reduced even in basal conditions [36].The maintenance phase, lasting one or two weeks, is characterized by a stabilized GFR at the lowest levels, with little or no urine production and possible uremic complications [36].The cells begin the repair process to restore the integrity of the tubular system through migratory and proliferative activity, with a slow improvement in cellular function. The reason for a low GFR when proceeding towards normalization of systemic hemodynamics is still unclear: among the proposed explanations are persistent intrarenal vasoconstriction and medullary ischemia, probably maintained by the release of vasoactive mediators from the endothelium damaged of the peritubular vessels and by a hyperactivity of the tubuloglomerular feedback [36].Finally, there is the recovery phase, in which the reparative processes are completed with normalization of the GFR. The tubular cells recover with a certain delay with respect to the normalization of the glomerular filtrate; for this reason, it is possible that a polyuria due to non-reabsorption of solutes is established.

The contrast medium form is the most common [37] and is characterized by a rapid peak within 48 h—but reversible within 3–5 days—of plasma creatinine and urea. It is due to the induction of intrarenal vasoconstriction and, to a varying extent, to direct toxic damage through the production of reactive oxygen species (ROS) and consumption of antioxidants. The probability of development is related to the dose and type of contrast medium (easier if hyposmolar), as well as to the patient’s condition: at-risk subjects with congestive heart failure, diabetes mellitus, and volume depletion.

As far as the drug-induced form is concerned, antibiotics, antivirals, and chemotherapeutics are most involved, through various mechanisms, such as the following [38]:Intrarenal vasoconstriction;Blocking oxidative phosphorylation;Precipitation in the tubules;The release of free radicals.

To name a few, the aminoglycoside gentamicin causes nephrotoxicity in more than 30% of patients treated for more than 7 days, particularly at high doses, through mitochondrial damage [39]. Cisplatin, widely used in the therapy of neoplasms causes tubular damage through an important inhibition of the cellular respiration [40]. The analgesics diclofenac and acetaminophen, the use of which is increasing for the treatment of arthritic forms, damage the kidney both due to the inhibition of prostaglandin synthesis and by direct action on the tubules with oxidative damage [41,42,43].

The postrenal form of AKI is the least represented of the three, and its origin is the obstruction of the urinary tract. This can be located at the level of the ureters, due to the presence of stones, clots, flaking material from the papilla, masses, or external compression, as in the case of retroperitoneal fibrosis; at the level of the bladder neck, in case of prostatic hypertrophy, neurogenic bladder, and tumors; and at the urethral level, due to the presence of congenital stenosis or valves. 

The onset of renal failure could be linked to surgical procedures before a serious impairment of organ function. The reduction of urinary production in the course of persistent increase in intra-abdominal pressure (IAP) is difficult to classify, a condition that can commonly occur in the course of ascites, hemoperitoneum, intestinal distension from occlusion or ileus, and surgical packing, and which often leads to organ compromise. It has in fact been seen that the higher the IAP is, the lower the urinary output [44,45]. A study [46] has shown that the stabilization of the flow rate through blood volume expansion does not correct the renal deficit, with the GFR remaining persistently reduced to a fifth of the normal level. The problem, therefore, may not lie in the collapse of the venous return to the heart.

The answer could be in a direct compression of the renal parenchyma, with an increase in resistance so as to prevent perfusion even in the case of preserved cardiac output. It would, therefore, be a sort of intrinsic AKI in which the renal microcirculation and tubules are affected

### 3.3. AKI in the Course of Sepsis

ARF in sepsis develops in 50% of patients and is related to a mortality in up to one-third of cases [8]. It is a complication that appears to be early in 64% of patients, i.e., within the first day [8]. The pathogenesis of the so-called septic AKI is complex. It is mainly considered to be a pathology of the renal macro-circulation that involves a global ischemia of the organ due to hypoperfusion, linked to systemic vasodilation and vasopermeability mediated by the release of inflammatory cytokines. However, recent evidence has shown the possible development of AKI in the presence of preserved or even increased renal blood flow (RBF) [46,47]. This seems to be linked to a redistribution of the renal microcirculation, with a consequent imbalance between the hyperperfused cortex and the medulla, an area with the greatest oxygen consumption that undergoes hypoxia and ischemia [48]. This microcirculatory dysfunction sets in hours before the onset of clinical AKI, leading to decreased renal function. Paradoxically, the use of vasopressors, by increasing the RBF, can worsen this imbalance, with overload of the nephrons at the cortical level and a consequent increase in the oxygen demand at the medullary level. Experimental studies have shown harmful effects at the level of renal microcirculation with the use of high doses of catecholamines, which is why a new line of research is moving towards the use of non-catecholaminergic vasopressors, such as vasopressin and angiotensin II, which preserve the oxygenation of the renal medulla and not only the RBF [46]. A second mechanism is instead linked to post-apoptotic necrosis induced by bacterial lipopolysaccharide (LPS), released during sepsis, capable of binding to membrane molecules of the body’s cells, including renal ones, and inducing apoptosis. When the release is massive, the ability of macrophages to phagocytize apoptotic bodies is overcome with degeneration of the same and production of pro-inflammatory molecules with triggering of necrosis [41,49,50].

### 3.4. Therapy

#### 3.4.1. RRT

The decision to initiate an RRT is often based on clinical aspects such as water overload and on biochemical parameters of metabolic and electrolyte imbalance: this is necessary in approximately 85% of patients presenting with AKI associated with oligoanuria and in 30% of non-oliguric forms [51,52]. However, in the absence of these factors, their initiation tends to be delayed as much as possible, due to the potential risks associated with the procedure, linked to vascular access, anticoagulation, hemodynamic alterations, and the fear of accelerating the progression towards CKD [50].

In ICU, the CRRT is the most appropriate, given the frequent hemodynamic instability; it allows for a trans-compartmental balance of the solutes and a removal of dilated fluids over 24 h, allowing for better tolerability:Slow continuous ultrafiltration (SCUF), in which only the removal of fluids occurs, used in patients with water overload;Continuous veno-venous hemofiltration (CVVH), in which convection is exploited with the removal of fluids and solutes and therefore the need for reintegration;Continuous veno-venous hemodialysis (CVVHD), in which countercurrent diffusion is used;Continuous veno-venous hemodiafiltration (CVVHDF), which combines the two previous modalities, with greater purifying efficacy.

Continuous high-flow veno-venous hemodialysis (CVVHFD) is a diffusive technique with high flows that result in even a minimal convective component, despite the fact that no replacement fluid is used.

#### 3.4.2. Non-Replacement Therapies [53,54]

Non-replacement therapy can be divided into a support form and a pathogenetic form, aimed at restoring the basal conditions to normality.

As for the first, this aims to prevent and/or treat complications such as hyperkalemia and acidosis, as well as hyperphosphatemia and hypermagnesemia.

Among the therapies aimed at blocking pathogenetic processes, an important role is played by hemodynamic support, with the restoration of an adequate volume and perfusion pressure, through the reintegration of fluids and amino support. Therapy with intrarenal vasodilators such as fenoldopam or atrial natriuretic peptides, although there are some encouraging studies on the improvement of renal function, currently does not have sufficient scientific evidence to be recommended. 

As for diuretics, furosemide acts on the kidney loop and also has a vasodilating action. It can reduce the metabolic work at the level of the ascending portion and contribute to the unblocking of the nephron by any cylinders and cellular debris. It is also able to reduce the concentration of toxins such as myoglobin and hemoglobin and can convert an oliguric ARF into the non-oliguric form, simplifying patient management, as fluid intake and nutrition are less conditioned. However, it does not necessarily affect the course of acute renal failure, it can worsen the forms of contrast media [36], and it is ototoxic in excessive doses.

Fenoldopam mesylate was found to be effective in reducing the onset of postoperative AKI when used before the development of the kidney damage. Positive results were also obtained in the management of intensive-care-unit patients with AKI, although the clinical studies investigated were few and conducted on small samples [55,56].

#### 3.4.3. Basic Principles of RRT [51]

The hemodialyzer is the extracorporeal circuit device that is suitable for purification, regardless of the treatment modality. In most cases, it turns out to be hollow fiber. Each fiber, cylindrical in shape, allows the transport of fluids and solutes through a semi-permeable porous surface, acting as a membrane. The water permeability of the membrane represents the ultrafiltration coefficient (Kuf), and it is obtained from the product of the hydraulic permeability (Lh) for the surface of the membrane (A) [13,27,53,54]:Kuf = Lh × A

It is measured by the manufacturer, expressed in mL/h/mmHg, depends on the porosity, and is the main element to define membranes with high (>25 mL/h/mmHg) and low flow (<10 mL/h/mmHg). The mass transport coefficient (K_0_) is the overall resistance that limits the diffusive transport of a solute across the entire membrane surface of a hemodialyzer, and it is expressed in mL/min. The Sieving coefficient (Cs) is specific for each solute and for each membrane, and it can be approximated as the ratio between the concentration of the solute in the ultrafiltrate or dialysate and in the plasma at the inlet of the hemofilter. It ranges from 0 to 1.
Cs = [solut] Ultrafiltrate/[solut] plasma
Cs = [solut] Dialisate/[solut] plasma

For drugs, it can also be expressed as a protein-binding free fraction (PB), assuming that this is totally filtered [13,16]:Cs = 1 − PB

For a membrane, the cutoff value represents the molecular weight (MW) of the smallest solute not affected by the transmembrane removal, and it essentially depends on the size of the pores. Generally, the MW of a solute characterized by a Cs of 0.1 is expressed as a 10% cutoff (0.1 cutoff). In a similar way, a 90% cutoff (cutoff 0.9) can be identified. High cutoff membranes have a Cs for albumin >0.

The transport of solutes occurs mainly by diffusion or ultrafiltration, and secondarily by absorption. The transport of fluids, on the other hand, can only take place by ultrafiltration. Ultrafiltration is guided by a pressure gradient between the blood compartment and the hemofilter, with the passage of fluids and, consequently, of solutes by convection. It is influenced by the intrinsic properties of the membrane, such as Kuf, and by the parameters set, such as the obtained trans membrane pressure (TMP). Ultrafiltration is defined by the ultrafiltration flow (Quf)
Quf = Kuf × TMP

Convection is expressed by the convective flow (Fc), which depends on the solute concentrations [solut], the Cs of the solute Quf:Fc = Quf × Cs × [solut].

Diffusion is the process by which the solute molecules move from an area with a higher concentration to one with a lower concentration through a semipermeable membrane, until equilibrium is reached. It is expressed by the diffusive flux (Fd) and depends on the temperature (T), the surface (A), the diffusivity coefficient (D), and inversely proportional to the thickness (S):Fd = A × T × D/S

The diffusivity coefficient turns out to be inversely proportional to the MW and to the size of the molecule. Absorption is an extracorporeal process in which blood and plasma compounds—in particular, peptides and proteins—bind to the limbs or other absorbent substances, such as gels, resins, or monoclonal antibodies. This occurs mostly at the pore level, so membranes with a more open pore structure, such as high fluxes, will potentially have higher absorption. The various modalities of RRT exploit different mechanisms for solute removal. In hemodialysis (HD), the main mechanism is diffusion, which is particularly effective in the clearance of small solutes. In the hemodialyzer, blood and a dialysis solution circulate against the current (less frequently co-current). In this way, the average concentration gradient is kept high along the entire length of the dialyzer.

Hemofiltration (HF) uses only ultrafiltration and therefore convection, with the infusion of a sterile solution into the blood circuit to replace the eliminated plasma volume. This can be performed prefilter (predilution) or postfilter (postdilution). The latter is more efficient because it does not dilute the solutes before they enter the filter, but it can be associated more easily with the clogging of the membrane for greater hemoconcentration. Hemodiafiltration (HDF) combines the previous methods. The transport of solutes during extracorporeal treatments depends on the flow of blood, the flow of the dialysis and replacement fluid, and the set ultrafiltration. Blood flow (QB) is the volume of blood circulating in the extracorporeal circuit per unit of time, usually expressed in mL/min. It depends on the type and quality of vascular access and the modality used. The replacement volume (VR) is the amount of fluid returned to the patient in pre- or postdilution or both. The ultrafiltrate volume (Vuf) is the total amount of liquid removed during a treatment by a positive TMP inside the hemofilter.

The ultrafiltration flow (Quf) is the amount of ultrafiltrate produced for each time interval, usually expressed in L/h or in mL/min. The filtration fraction (FF) is the ratio of ultrafiltration flow to blood flow, and it should be kept below 30% to reduce hemoconcentration and protein–membrane interaction, especially if predilution is not used.
FF = Quf/QB

The dialysate volume (VD) is the amount of dialysis fluid that flowed into the hemodiafilter during the entire treatment; the dialysate flow is the amount of fluid per unit of time, usually L/h or mL/min. The effluent volume (Veff) is the volume of waste fluids coming from the outlet port of the dialysate-ultrafiltered compartment; the effluent flow (Qeff) is its expression per unit of time. The dose is the amount of blood purified from waste products during an RRT treatment and is measured as the removal flow of a representative solute, usually urea. It is expressed in mL/kg/h. Efficiency is identified with the concept of clearance and represents the volume of blood purified from the solute during a time interval. The intensity of the treatment is given by the efficiency for the time of administration.

Regarding the clearance (Cl) of drugs, in the case of hemofiltration with reinfusion in postdilution, this will be given by the product of the Sieving coefficient (Cs) and the ultrafiltration flow (Quf):Clpost = Qf × Cs

Meanwhile, in predilution, it will be reduced due to the presence of a dilution factor (Df), given by the ratio between the blood flow (Qb) and the sum between the blood flow and the return volume (Vr):Clpre = Qf × Cs × Df
Df = Qb/Qb + Vr

The clearance during hemodialysis will be given by the dialysate flow (QD) for the Sieving coefficient of the drug:Cld = QD × Cs

If both methods are used, clearance will result from the sum of the ultrafiltration and dialysis flow times the Sieving coefficient:Clfd = (Qf + QD) × Cs

## 4. Antibiotics in CRRT

As previously mentioned, AKI is an easily encountered condition in intensive care, with a high mortality rate and a greater frequency in the course of sepsis. The need for CRRT is increasing, also because it ensures a better outcome the earlier it is applied. Optimal antibiotic therapy is essential in these patients, but the drug elimination rate during CRRT can be highly variable, depending on the method used, the characteristics of the same, and the patient’s condition. Sepsis itself increases Vd, prolongs t_1/2_, and alters the protein binding of many antibiotics, with increased capillary permeability due to the release of inflammatory mediators with fluid accumulation and hypoalbuminemia [55,56]. For example, the Vd of aminoglycosides increases by about 25% in the critically ill patient, while the Vd of beta-lactams and of Vancomycin changes less but with important individual variations [57,58,59]. Table 4 shows main factors conditioning the removal of antibiotics in CRRT.

Antibiotics with a low volume of distribution (<1 L/kg) will be more affected by removal during CRRT than those with a high Vd (>2 L/kg), especially in the course of slow removal techniques, i.e., the main ones used in intensive care, due to the possibility of a continuous redistribution of the drug from the tissues to the blood [58]. The parameters set on the machine can modify clearance in various ways: the increase in blood flow or dialysate can increase the transmembrane pressure and, therefore, the removal of antibiotics [60]. Although the KDIGO guidelines recommend an effluent dose of 20–25 mL/kg/h, it has been found that the initial prescribed dose is often higher [55,60]. With this, the risk of being sub-therapeutic at the standard dosage of antibiotic occurs, thus calling for higher dosages [60,61,62]. The choice to replace the fluids removed in predilution or in postdilution will have an impact on drug clearance, reducing it in the first case and enhancing it in the second. Furthermore, the use of biosynthetic membranes with larger pores than conventional filters involves the removal of drugs with a higher molecular weight [58,59,62,63]. Finally, the duration of use of the circuit may also have a role in the removal of drugs. In fact, there is a tendency for the formation of a second membrane on the filter for protein deposition from the plasma during CRRT, with a reduction in transmembrane clearance and therefore in the performance of the filter [64].

Due to the lack of defined guidelines on antibiotic dosage in CRRT, many equation-based models (Table 5), often complicated and unsupported by clinical data, have been proposed [60,65,66,67,68]. In fact, the ongoing clearance of CRRT is significant when this represents more than 25–30% of the total body clearance for a given drug, but for many of them, the clearance is estimated and not measured [58]. Furthermore, the Sieving coefficient, which determines how permeable the filter membrane is to the drug, is often calculated based on the protein binding of the drug, obtained from tables based on data in healthy volunteers, not reflecting the conditions present in the critically ill patient [55].

The ongoing clearance of CRRT (ClCRRT) is considered relevant for drugs with predominantly renal elimination, with a reduced Vd and low protein binding, and therefore a significant Sieving coefficient [55]. As for antibiotics, therefore, it should be hydrophilic, in particular. However, there are exceptions: for example, among the Beta-lactams, Ceftriaxone and Oxacillin have mainly biliary elimination; among the Fluoroquinolones, Levofloxacin and Ciprofloxacin, despite their lipophilicity, have renal clearance [56]. Therefore, it may be necessary to increase the doses compared to the patient in renal insufficiency without replacement therapy, although we tend to consider the patient in CRRT with a reduced clearance, considering the GFR between 10 and 30 mL/min [56]. From this, it is clear that it is not possible to identify a standard dosage on this assumption, because there is a risk of oversimplification, thus failing the therapeutic goal. In fact, it has been seen that, in 25% of cases, the antibiotic therapy does not reach the necessary concentration target. Moreover in 15% of cases, there is no achievement of the MIC and in 40%, the drug concentration is above the considered optimal MIC [57,69,70,71].

The optimal dosing regimen of vancomycin for critically ill patients receiving continuous venovenous hemofiltration (CVVH) remains controversial, not to mention those with concurrent use of extracorporeal membrane oxygenation (ECMO).

Wang et al., conducted a study based on a Monte Carlo simulation (MCS) to assess the proper optimization of vancomycin dosage regimens in CVVH patients. The population typical vancomycin clearance (CL) was 1.15 L/h, and the median volume of distribution was 16.9 L. The CL was significantly correlated with the ultrafiltration rate and albumin level. They suggested the following regimen doses: (i) 5 mg/kg q8h for patients with normal albumin levels and hypoalbuminemia but with ultrafiltrate rate between 20 and 25 mL/kg/h; (ii) 10 mg/kg qd in the case of a normal level of albumin and ultrafiltrate rate between 20 and 35 mL/kg/h; and (iii) 10 mg/kg q12h in the case of hypoalbuminemia and an ultrafiltrate rate between 25 and 40 mL/kg/h [72].

Li et al., conducted a study to determine the Pk and maintenance dose of vancomycin in severe pneumonia in 10 patients receiving CVVH performed in mixed predilution and postdilution mode. Group A received an initial dose of 500 mg only, whereas Group B received 500 mg every 12 h until steady state was achieved. Serum and ultrafiltrate were collected over 12 h after infusion of vancomycin. After initial dosing, the mean sieving coefficient (SC) was 0.72 ± 0.02, and CVVH clearance (CLCVVH, 1.35 ± 0.03 L/h) constituted 60.55% ± 13.69% of total vancomycin clearance (CLtot, 2.36 ± 0.72 L/h). When steady state was reached, the SC of the patients was 0.71 ± 0.03, and the CLCVVH (1.34 ± 0.06 L/h) accounted for 66.96% ± 6.05% of the (total clearance) CLtot (2.03 ± 0.27 L/h). They concluded that therapeutic vancomycin levels are difficult to be maintained in patients on CVVH and suggested (i) close monitoring of serum trough concentrations and (ii) a maintenance dose of 400–650 mg every 12 h calculated on CLtot to achieve a trough concentration of 15–20 mg/L at steady state [73].

Wahby et al., performed a retrospective cohort study to evaluate current dosing practices for patients treated with CVVH and develop guidelines for the optimal dosing and monitoring of vancomycin to improve target trough attainment. Patients were included if they received vancomycin during CVVH for at least 48 h. There were 141 patients with 443 random vancomycin serum levels in the DBL group and 59 patients with 143 vancomycin trough levels in the SD group. Mean vancomycin trough levels were similar between groups (17.1 ± 6 vs. 16.5 ± 4 mcg/mL) for the DBL and SD groups, respectively. For the primary end point of overall target trough achievement of 15–20 mcg/mL, significantly more trough levels in the SD group were in the 15–20 mcg/mL range compared with the DBL group, 50% vs. 38% (*p* < 0.001), respectively. When the target trough range was extended to 10–20 mcg/mL, the success rates were similar between groups (74% DBL vs. 82% SD, *p* = 0.021). Scheduled vancomycin dosing regimens of 15–22 mg/kg every 12–24 h were required to yield trough levels in the 15–20 mcg/mL range. They concluded that target vancomycin trough achievement of 15–20 mcg/mL occurred more frequently when vancomycin was scheduled at a dose of 15–22 mg/kg every 12–24 h based on the ultrafiltration rate and may alleviate the time and cost associated with frequent vancomycin serum monitoring [74].

Sin et al., proposed a prospective observational study of patients treated with intravenous infusion (CIV) regimen in patients undergoing CVVH that incorporates weight-based CVVH intensity (mL/kg/h) into the dosing nomogram. The primary outcome was the achievement of a therapeutic vancomycin concentration (15–25 mg/L) at 24 h. Secondary outcomes included the achievement of therapeutic concentrations at 48 and 72 h. The nomogram was analyzed in 52 critically ill adults. Vancomycin concentrations were therapeutic in 43/52 patients (82.7%) at 24 h. Of the nine patients who were not therapeutic at 24 h, seven were supratherapeutic and two were subtherapeutic. The mean (SD) concentration was 20.1 (4.2) mg/L at 24 h, 20.7 (3.7) mg/L at 48 h, and 21.9 (3.5) mg/L at 72 h. Patients with a CVVH intensity >20 mL/kg/h experienced higher CLvanc at 24 h compared with patients with CVVH intensity <20 mL/kg/h (3.1 versus 2.6 L/h; *p* = 0.013). They found that the majority of patients achieved therapeutic concentrations at 24 h and maintained them within range at 48 and 72 h [75].

Yang et al., aimed to determine if a new dosing regimen could achieve the target vancomycin trough concentration (Ctrough) of 10–20 mcg/mL in patients receiving CVVH. The vancomycin dosing regimen was 15–20 mg/kg as the loading dose and 7.5 mg/kg every 12 h as the maintenance dose. Serum concentration was determined after at least four doses of vancomycin were given. A total of 17 patients were enrolled. The ultrafiltration rate of CVVH was 30.6 ± 5.5 mL/kg/h with the Ctrough of 14.7 ± 3.5 mcg/mL. They concluded that all patients receiving CVVH achieved the target Ctrough with this new dosing regimen [76].

Xu et al., performed a prospective interventional trial to compare the Pk and Pd target attainment, therapeutic efficacy, and safety among critically ill patients who received CIV or (intermittent) II infusion of vancomycin and to explore the correlations of effluent flow rate (EFR) with Pk/Pd indices. The primary outcome was to compare the Pk/Pd target attainment, including target concentration and target area under the curve over 24 h to minimum inhibitory concentration (AUC24/MIC). The overall target attainment of Pk/Pd indices was higher with CI, as compared to II, irrespective of target concentration (78.7% vs. 40.5%; *p* < 0.05) or AUC24/MIC (53.2% vs. 28.6%; *p* < 0.05). There were no significant differences in the clinical success (72.2% vs. 50.0%; *p* = 0.183) and microbiological success (83.3% vs. 75.0%, *p* = 0.681) between the patients treated with CI or II of vancomycin. Adverse reactions occurred at similar rates (0.0% vs. 4.4%; *p* = 0.462), and the mortality between the two modalities was also not significantly different (21.7% vs. 17.9%; *p* = 0.728). The correlation analysis showed a weak to moderately inverse correlation of EFR with observed concentration (r = −0.3921, *p* = 0.01) and AUC24/MIC (r = −0.3811, *p* = 0.013) in the II group, whereas the correlation between EFR and observed concentration (r = −0.5711, *p* < 0.001) or AUC24/MIC (r = −0.5458, *p* < 0.001) in the CI group was stronger. CI of vancomycin in critically ill patients undergoing CVVH was associated with improved attainment of Pk/Pd indices. Furthermore, the inverse correlation of Pk/Pd indices with EFR was stronger among patients treated with CI of vancomycin [77].

Zheng et al., performed a prospective study comparing the Pk and Pd of linezolid in patients with sepsis receiving CVVH or extended daily hemofiltration (EDH). Pk and Pd were analyzed by using MCS. From 20 patients, 320 blood samples were collected for Pk and Pd analysis. Pk profiles of linezolid were best described by a two-compartment model. Pk parameters were not significantly different between EDH and CVVH groups and were associated with body weight, renal replacement therapy (RRT) duration, and sequential organ failure assessment score. MCS showed poor fractional target attainment for a MIC of 2 mg/L with standard 600 mg intravenous administration every 12 h. Patients with sepsis receiving RRT exhibited variability in Pk/Pd parameters for linezolid. The Pk parameters were not significantly different between CVVH- and EDH-treated patients. A higher probability of target attainment would be achievable at a MIC of 2 mg/L in EDH patients. Higher linezolid doses should be considered for patients on RRT to achieve adequate blood levels [78].

Altered pharmacokinetics (Pk) of hydrophilic antibiotics in critically ill patients is common, with possible consequences for efficacy and resistance. Werumeus Buning et al., aimed to describe ceftazidime population Pk in critically ill patients with a proven or suspected *Pseudomonas aeruginosa* infection and to establish optimal dosing. A population Pk model was constructed, and probability of target attainment (PTA) was assessed for targets 100% T > MIC and 100% T > 4 × MIC in the first 24 h. Ninety-six patients yielded 368 ceftazidime concentrations. Variability in ceftazidime clearance (CL) showed association with CVVH. For patients not receiving CVVH, variability in ceftazidime CL was 103.4% and showed positive associations with creatinine clearance and with the comorbidities. Patients receiving loading doses before continuous infusion demonstrated higher PTA than patients who did not (100% T > MIC: 95% (*n* = 65) vs. 13% (*n* = 15); *p* < 0.001 and 100% T > 4 × MIC: 20% vs. 0%; *p* = 0.058). The considerable variability in ceftazidime Pk in ICU patients could largely be explained by renal function, CVVH use, and several comorbidities. Critically ill patients are at risk for underexposure to ceftazidime when empirically aiming for the breakpoint MIC for *Pseudomonas aeruginosa*. A loading dose is recommended [79].

Sember et al., built Pk models, using published pharmacokinetic/demographic data, to predict drug disposition in 5000 virtual critically ill patients receiving CVVH with the standard (20–30 mL/kg/h) and a higher (40 mL/kg/h) effluent rate. MCS was performed to assess the probability of target attainment (PTA) of four cefepime and ceftazidime doses administered over 4 h, with the target of ≥60% free T > 4 × MIC. The lowest dose attaining PTA ≥90% during the first 48 h was considered optimal. Cefepime 2 g loading dose (LD), then an extended infusion of 2 g every 8 h was optimal in CVVH at 20 mL/kg/h and the same ceftazidime dose was optimal in CVVH at 20–30 mL/kg/h. Higher cefepime and ceftazidime doses were required to be optimal at higher effluent rates. This optimal dose, particularly for cefepime, likely increases the neurotoxicity risk in most virtual patients with all CVVH settings. The authors found that cefepime and ceftazidime 2 g LD, followed by the extended infusion of 2 g every 8 h, may be optimal in CVVH with standard effluent rates [80].

Philpott et al., conducted a prospective open-label Pk study in which ten critically ill adults received extended infusion (EI) of cefepime 2 g intravenously every 8 h as a 4 h infusion while receiving CVVH(D). Concurrent serum and CRRT effluent samples were collected at hours 1, 2, 3, 4, and 8 after the first cefepime dose and after either the fourth, fifth, or sixth (steady-state) cefepime dose. The Pk analyses included CRRT clearance, half-life, and sieving coefficient or saturation coefficient. The cefepime peak (4 h) concentrations, trough (8 h) concentrations (Cmin), and minimum inhibitory concentration breakpoint of 8 µg/mL for the pathogen (MIC) were used to evaluate attainment of pharmacodynamic targets: 100% of the dosing interval that free drug remains above MIC-8. The total CRRT effluent flow rate was the mean ± SD of 30.1 ± 5.4 mL/kg/h, CRRT clearance was 39.6 ± 9.9 mL/min, and half-life was 5.3 ± 1.7 h. The SC and saturation coefficient were 0.83 ± 0.13 and 0.69 ± 0.22, respectively. The first and steady-state dose Cmin were 23.4 ± 10.1 µg/mL and 45.2 ± 14.6 µg/mL, respectively. They assessed that no significant differences were observed in Pk properties between the first and steady-state doses among or between patients. It may be reasonable to initiate an empiric or definitive regimen of EI cefepime in critically ill patients receiving concurrent CRRT who are at risk for resistant organisms. Further research is needed to identify the optimal dosing regimen of EI cefepime in this patient population [81].

Por et al., aimed to compare imipenem clearance (CL) in burn patients with and without CVVH, determine the effect of burn on imipenem volume of distribution (CVVH, n = 12; no CVVH, n = 11), in combination with previously published models. MCS were conducted to evaluate the probability of target attainment. A two-compartment model best described the data. They provide direct comparison of imipenem CL in burn patients with and without CVVH. Notably, there was no significant difference. Large imipenem Vd in patients with severe burns is likely explained by increased capillary permeability, for which serum albumin may be a reasonable surrogate. Dosing 500 mg every 6 h is adequate for burn patients on renally dosed CVVH; however, suspicion of augmented renal clearance or patients placed on CVVH without renal impairment may necessitate dosing of 1000 mg every 6 h [82].

Selig et al., evaluated the Pk parameters of meropenem from 23 critically ill patients, burn or non-burn, treated with or without CVVH to determine the contribution of burn and CVVH to the variability of therapeutic meropenem levels. They applied a two-compartment model that best described the data and revealed the creatinine clearance (CrCl) and total burn surface area to be significant covariates on clearance (CL) and peripheral volume of distribution (Vp), respectively. Of interest, non-burn patients on CVVH displayed an overall lower inherent CL as compared to burn patients on CVVH (6.43 vs. 12.85 L/h). Conclusions assessed a standard dose of 1000 mg every 8 h; however, if ARC is suspected, or the severity of illness requires a more stringent therapeutic target, we recommend a loading dose of 1000–2000 mg infused over 30 min to 1 h, followed by continuous infusion (3000–6000 mg over 24 h) or intermittent infusion of 2000 mg every 8 h [83].

Jang et al., performed a study to predict the optimal ceftazidime, cefepime, meropenem, and piperacillin/tazobactam doses in patients undergoing CVVH. An MCS was performed by using published Asian demographics and Pk parameters in 5000 virtual patients at three CVVH effluent rates (Qeff; 20, 30, and 40 mL/kg/h). Ceftazidime 1 g q12h, meropenem 1 g q12h, and piperacillin/tazobactam 3.375 g q6h were optimal for all Qeff settings against fT > 1 × MIC. Cefepime 2 g q24h and 2 g q12h were optimal at 20 and 30–40 mL/kg/h, respectively. For the aggressive Pd target (4 × MIC), optimal ceftazidime regimens were 1.25 g q8h (20–30 mL/kg/h) and 1.5 g q8h (40 mL/kg/h). Cefepime 2 g q8h and meropenem 1 g q8h were optimal at all Qeff settings. No simulated piperacillin doses attained the aggressive Pd target. They found that MCS enabled the prediction of optimal β-lactam dosing regimens for Asian patients receiving CVVH at varying Qeff settings [84].

Körtge et al., studied the effect of CVVH on concentrations of antibiotics with low (meropenem), medium (vancomycin), and high (daptomycin) protein binding (PB). Clearances and sieving coefficients (SCs) were determined from antibiotics concentrations measured at filter inlet, outlet, and filtrate side. Reservoir concentration data were fitted by using a first-order kinetic model. Meropenem and vancomycin concentrations decreased to 5–10% of the initial plasma level, while only 50% of daptomycin was removed. Clearances and SCs were (10.8 [10.8–17.4] mL/min, SC = 0.72 [0.72–1.16]) for meropenem, (13.4 [12.3–13.7] mL/min, 0.89 [0.82–0.92]) for vancomycin, and (2.1 [1.8–2.1] mL/min, 0.14 [0.12–0.14]) for daptomycin. Removal by adsorption was negligible. The clearances and SCs presented are comparable with findings of other authors. Meropenem and vancomycin, which exhibit low and medium PB, respectively, were strongly removed, while considerably less daptomycin was removed because of its high PB [85].

Bellmann et al., studied levofloxacin clearance in critically ill patients with impaired renal function on CVVH; mean half-life was prolonged by a factor of about 3 (20–25 h). They determined a wide variability in Pk. The half-life was about 30 h, and the mean levofloxacin clearance was raised by a factor of 2. The area under the concentration–time curve was reduced by hemofiltration, while the volume of distribution was increased. There was a positive correlation between blood flow through the hemofilter and levofloxacin clearance. Variable amounts of the drug were recovered from the hemofilter. Most plasma levels, however, were in the therapeutic range, and drug accumulation to toxic plasma concentrations was not observed in renal failure patients undergoing CVVH and receiving single daily administration of 0.5 g of levofloxacin i.v. During CVVH, using polysulfone membrane hemofilters, plasma concentrations of levofloxacin are not easily predictable [86].

Wang et al., analyzed the population Pk of polymyxin B in patients receiving CVVH to optimize individual dosing regimens in specific clinical scenarios. The population Pk analysis and MCS were performed. The AUC-24 at steady state of polymyxin B during CVVH was 27.94 ± 10.92 mg‧h/L, which is significantly lower than that outside CVVH (77.89 ± 35.66 mg‧h/L) (*p* = 1.65 × 10^−8^). The population pharmacokinetic model revealed that CVVH significantly increased the clearance of polymyxin B. Monte Carlo simulations showed that, for patients on CVVH, a loading dose of 200 mg plus a fixed maintenance dose of 150 mg every 12 h had a high probability of achieving AUCss, 24 h of 50–100 mg‧h/L and the pharmacokinetic/pharmacodynamic target with a minimum inhibitory concentration ≤0.5 mg/L. They established that, for patients undergoing CVVH, high doses of polymyxin B and a dose-adjustment regimen based on therapeutic drug monitoring should be considered to improve efficacy [87].

## 5. Methods

The present review was based on the following research question: what is known about the antibiotics posology and clearance in critically ill with renal failure in RRT. The relevance of such a question depends on the RRT device filter clearance and on antibiotics steric hindrance. For the aims of the present review, Selection of Literature Primary original research articles published in peer-reviewed journals were collected from the PubMed database between August 2022 and January 2000 by using all of the following keywords organized in 4 main domains: (1) Domain 1, critically ill patient(s) OR intensive care unit OR ICU; (2) Domain 2, study type—study OR trial; (3) Domain 3, renal failure or kidney failure; Domain 4, renal replacement therapy OR continuous renal replacement therapy; and Domain 5, antimicrobial(s) AND/OR sepsis. The 5 domains were combined by the AND operator, and only full articles in English were retrieved during the first round of the literature search. The nature of the review is “narrative”.

## 6. Conclusions

The increase in infections associated with high disease severity and parallel antibiotic resistance has made the timely initiation of adequate antibiotic therapy even more necessary. For the patient in acute renal insufficiency, there are shared guidelines borrowed from the literature data, and above all, there is a certain need to reduce the dosage in relation to the reduction in creatinine clearance, especially when this is reduced to <30 mL/min. However, concerning CRRT patients, there are few data that can inform daily clinical practice.

Apart from many equation-based models, which are often complicated and only theoretical, without support from clinical data, the lack of sufficient data in the literature and guidelines does not allow us to set up correct antibiotic therapy adjustments in the critically ill in RRT; therefore, we must take into account the following considerations:

The therapeutic posology should be increased both as a loading dose and as a maintenance dose by evaluating the interaction and presence of the following variables, even though there is no indication to quantitatively inform the dose adjustment. The loading and maintenance doses of the antibiotic must take into account the following variables:Pk/Pd characteristics of antibiotics:Vd: Antibiotics with a low volume of distribution (<1 L/kg) will be more affected by removal during CRRT than those with a high Vd (>2 L/kg);Protein binding (the higher the amount of the antibiotic bound to albumin, the more renal elimination is reduced) is essentially linked to the albumin level.The CRRT methodology (CVVH/CVVHD/CVVHDF):It impacts on the renal purification and clearance of the drug, making the drug-elimination rate extremely variable; the most pragmatic advice is to keep the CVVH(D)(F) profile as stable as possible;Moreover, sepsis itself increases Vd, prolongs t1/2, and alters the protein binding of many antibiotics due to increased capillary permeability induced by the release of inflammatory mediators with fluid accumulation and hypoalbuminemiaType of cartridges:The use of cartridges provided by biosynthetic membranes, with larger pores than conventional filters, involves the removal of drugs with a higher molecular weight;The duration of use of the circuit may also have a role in the removal of drugs. In fact, there is a tendency for the formation of a second membrane on the filter for protein deposition from the plasma during CRRT, with a reduction in the cartridge transmembrane clearance and, therefore, performance.

## Figures and Tables

**Table 1 antibiotics-11-01769-t001:** Types of antibiotics according to the type of bacterial inhibition.

Bacteriostatic Antibiotics	Bactericide Antibiotics
Tetracyclines	Penicillins beta-lactams inhibitor
Macrolides	Cephalosporins
Streptogramins	Carbapenems—monobactams
Sulfonamides/Trimethoprim	Quinolones
Lincosamides	Glycopeptides
Nitrofurans	Lipopeptides
Chloramphenicol	Lipoglycopeptides
Fusidic acid	Oxazolidinones
	Polymyxins
	Aminoglycosides
	Rifamycins
	Metronidazole

**Table 2 antibiotics-11-01769-t002:** Most-used and -studied time- and concentration-dependent antibiotics.

Time > MIC-Dependent	AUC > MIC-Dependent	C_max_ > MIC-Dependent
Beta-lactams	Glycopeptides	Aminoglycosides
Penicillins	Vencomycin	Gentamycin
Cephalosporin	Teicoplanin	Tobramycin
Carbapenems		Amikacin
Monobactams		Plazomycin
(including cefidecorol)		
Macrolides	Lipopeptides	Colistin
Erythromycin	Dalbavancin	
Chlarithromycin	Oratavancin	
Azitromycin	Telavancin	
Oxazolidinones	Tetracyclin	Metronidazole
Linezolid		
Tedizolid		
	Cycline derivative	Colistin
	Tigecycline	
	Eracacycline	
	Omadacyclin	
	Quinolones	
	Levofloxacine	
	Ciprofloxacin	
	Delafloxacin	
	Lincosamid	Lipopeptides
	Clyndamycin	Daptomycin

**Table 3 antibiotics-11-01769-t003:** Properties and Pk/Pd targets of the main antimicrobial drugs.

Antimicrobials	Pd Properties (Time/Concentration-Dependent Drugs)	Pk/Pd Target (Ratio or mg/dL)
Aminoglycosides		
Amikacin, Gentamicin, Isepamicin, Neomicin, Kanamycin, Paromomycin, Tobramycin, Netilmicin, Spectinomycin, Sisomicin, Dibekacin	Concentration-dependent	C_max_; MIC > 10
Macrolides		
Azithromicin, Chlarithromicin, Erythromycin	Time-dependent	C_min_ > 2
β-lactams		
*Penicillin and ß-lactam inhibitors*		
Ampicillin/sulbactam	Time-dependent	C_min_ > 8
Piperacillin/tazobactam	Time-dependent	C_min_ > 16/C_min_ > 4
*Cephalosporins*		C_min_ > 8
Cefazolin, Cefepime, Ceftriaxone	Time-dependent	C_min_ > 4–8 and or %T > MIC
Ceftazidime		
Carbapenems	Time-dependent	C_min_ > 4 and or %T > MIC
Meropenem;		
Imipenem-cilastatin		
Quinolones		
Nalidixic Acid, Norfloxacin, Ciprofloxacin, Levofloxacin, Gatifloxacin, Moxifloxacina, Gemifloxacin, Delafloxacin	Concentration-dependent	AUC:MIC > 100
Clindamycin	Time-dependent	C_min_ > 0.5
Colistin	Concentration-dependent	C_min_ > 4
Doxicyclin	Time-dependent	C_min_ > 4
Tetracyclines, Tigecyclin, Chloramphenicol	Time-dependent	C_min_ > 2
Metronidazole	Concentration-dependent	C_min_ > 4
Rifamycins	Concentration-dependent	C_min_ > 1
Rifampin, Rifabutin, Rifapentine, Rifamixin		
Trimetoprim/Sulfametoxazole	Time-dependent	C_min_ > 2/C_min_ > 38
Glycopeptides		
Vancoamycin	Concentration-dependent	AUC:MIC > 400
Teicoplanin	Time-dependent	(*Staphylococcus aureus*)
		C_min_ > 10–20
Lipopeptides		
Daptomycin	Concentration-dependent	AUC:MIC (not defined)
		C_min_ < 24.3 (less risk of myopathy)
Oxazolidinones		
Linezolid, Tedizolid	Concentration-dependent	C_min_ > 4 e/o %T > MIC
Streptogramins		
Quinupristin–Dalfopristin	Concentration-dependent	AUC:MIC (not defined)

**Table 4 antibiotics-11-01769-t004:** Factors modifying the removal of antibiotics in CRRT.

Factors	Comments
Pharmacokinetics	
Residual renal function	It adds to the clearance of the CRRT.
Non-renal clearance	May increase during AKI but decrease if concomitant hepatic insult.
Vd	If increased, less efficacy of clearance in CRRT; need for a higher loading dose.
Protein binding	Only the unbound fraction is cleared by the CRRT; clearance is increased if hypoalbuminemia.
Related to the CRRT	
Type of CRRT	Variations based on duration and type of solute removal, i.e., by diffusion, ultrafiltration, or both.
CRRT dose prescribed	The effluent volume is the main parameter that influences the elimination of drugs.
Blood flow	Little impact on the elimination of drugs if it is kept within the limits of usual prescriptions
Type of filter	It can affect the Sieving coefficient.
Filter surface	Impact not significant.

**Table 5 antibiotics-11-01769-t005:** Methods for estimating antibiotic dosage in CRRT.

Methods	Authors	CRRT	Equations	Assumptions
1	Golper et al. [16]	CVVH	D = C_ss_ × UBF × UFR × I	Dosage of antibiotic concentrations; Sieving coefficient corresponding to the unbound fraction of the drug.
2	Bugge et al [58]	CVVHDF	${D=D}_{N}\left(P_{x}+\left(1-P_{x}\right)\frac{{\mathrm{Cl}}_{\mathrm{CRtot}}}{{\mathrm{Cl}}_{\mathrm{CRn}}}\right)$	Dosage of antibiotic concentrations; Sieving coefficient corresponding to the unbound fraction of the drug; The normal dose of the drug is sufficient for optimal action.
3	Schetz et al. [66]	CVVH	${D=D}_{N}$	The normal dose of the drug is sufficient for optimal action.
4	Schetz et al. [66]	All types	$D=\frac{D_{\mathrm{anuria}}}{1-\left(\frac{{\mathrm{Cl}}_{\mathrm{EC}}}{{\mathrm{Cl}}_{\mathrm{EC}}{+\mathrm{Cl}}_{\mathrm{NR}}{+\mathrm{Cl}}_{R}}\right)}$	Dosage of antibiotic concentrations; The drug dose in anuric patients is sufficient for optimal action.

Legend: C_ss_, blood concentration at steady state; Cl_anur_, clearance of the drug in anurics; Cl_CRn_, normal creatinine clearance; Cl_Rtot_, renal and extracorporeal clearance; Cl_EC_, extracorporeal clearance; Cl_N_, normal total clearance; Cl_NR_, non-renal clearance; Cl_R_, renal clearance; D_anuria_, recommended dose in anuric patients; D_N_, recommended dose in patients with normal renal function; I, interval between doses; Px, fraction of extrarenal clearance (=Cl_ANUR_/C_lN_); Cs, Sieving coefficient; UBF, unbound fraction; UFR, ultrafiltration rate.

## Data Availability

Data is contained in Pubmed, Cochrane Databases.

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
