# Peer review of "Antibiotic Therapy in the Critically Ill with Acute Renal Failure and Renal Replacement Therapy: A Narrative Review"

_antibiotics, 2022, doi:10.3390/antibiotics11121769_

Round 1

Reviewer 1 Report

The topic is good and will contribute to the research and beneficial to the scientific community. However, the article needs to be written in depth and would be better to consider the below comments for improvements and mentioned the relevant things as per title of the article:

Needs to check the format and subtitle of the article as per journal guidelines.

1.    Abstract:

The abstract of 211 words is provided – it would be better if abstracts reflect the whole process/methods, objective and conclusion of the present study. The current layout is not truly reflecting the present study.

2.    Introduction:

Needs to mention very clearly the current gaps and what is the rationale for doing this present study.

3.    Results:

There are many things discussed under this subtitle and deviating from the title of the article.

Table 1 : Better to use “Bactericidal Antibiotics” word instead of “Bactericide Antibiotics”.

4.    Acute renal failure:

There are many things discussed under this subtitle and deviating from the title of the article.

5.    Conclusions

Needs to write in a precise way.

6.    References: The references not cited serially. Needs to cite and arrange all the references serially.  

Also, it would be better, if cite some more relevant newer references (last five years 2017-2022).

Line 82 and 83 – Needs to revise the reference citation - Only antibiotics that act on fundamental structures for 82 the bacterial cell, such as the wall or nucleic acids, will be bactericidal. [3,4,5] [3-5]

Vincent JL, Bihari D, Suter PM, et al. EPIC international Advisory Committee. The prevalence of nosocomial infection in inten-946 sive care units in Europe: results of the European Prevalence of Infection in Intensive Care (EPIC) study. JAMA. 1995;274(8):639-644

Also needs page number of reference number 3, 4, 8, 9, 10, 11, 14, 18, 25, 28, 31, 34, 35, 36, 37, 38, 39, 40, 42, 43, 44, 45, 46, 47, 48, 49, 50, 53, 57. 

Author Response

DEAR REVIEWER, THANKS FOR PROVIDING YOUR CONSIDERATIONS.

MUCH APPRECIATED, AS YOU HELPED ME TO MAKE MY MANUSCRIPT BETTER

I WILL ANSWER IN CAPITAL LETTER

MODIFIED TEXT IS HIGHLIGHTED IN YELLOW

1.    Abstract:

The abstract of 211 words is provided – it would be better if abstracts reflect the whole process/methods, objective and conclusion of the present study. The current layout is not truly reflecting the present study.

I HAVE SIGNIFICANTLY MODIFIED ABSTRACT

2.    Introduction:

Needs to mention very clearly the current gaps and what is the rationale for doing this present study.

I HAVE PROVIDED AS YOU ASKED

3.    Results:

There are many things discussed under this subtitle and deviating from the title of the article.

Table 1 : Better to use “Bactericidal Antibiotics” word instead of “Bactericide Antibiotics”.

RE-WRITTEN AND REDUCED ALL THE PARTS

CORRECTED THE ERRORS

4.    Acute renal failure:

There are many things discussed under this subtitle and deviating from the title of the article.

RE-WRITTEN AND REDUCED ALL THE PARTS

5.    Conclusions

Needs to write in a precise way.

RE-WRITTEN AND REDUCED ALL THE PARTS, HIGHLIGHTING TAKING HOME MESSAGE

6.    References: The references not cited serially. Needs to cite and arrange all the references serially.  

Also, it would be better, if cite some more relevant newer references (last five years 2017-2022).

Line 82 and 83 – Needs to revise the reference citation - Only antibiotics that act on fundamental structures for 82 the bacterial cell, such as the wall or nucleic acids, will be bactericidal. [3,4,5] [3-5]

Vincent JL, Bihari D, Suter PM, et al. EPIC international Advisory Committee. The prevalence of nosocomial infection in inten-946 sive care units in Europe: results of the European Prevalence of Infection in Intensive Care (EPIC) study. JAMA. 1995;274(8):639-644

Also needs page number of reference number 3, 4, 8, 9, 10, 11, 14, 18, 25, 28, 31, 34, 35, 36, 37, 38, 39, 40, 42, 43, 44, 45, 46, 47, 48, 49, 50, 53, 57. 

MODIFIED AS REQUESTED; THERE ARE 16 NEW MANUSCRIPTS ADDED IN THE REFERENCES AND DISCUSSED IN THE TEXT

Reviewer 2 Report

The authors aimed, in a narrative review to focus about what is known in the antibiotics posology and clearance in critically ill with renal failure in RRT. The relevance of such a question depends on the RRT device filter clearance and on antibiotics  steric hindrance.

The study covers some issues that have been overlooked in other similar topics. The structure of the manuscript appears adequate and well divided in the sections. Moreover, the study is easy to follow, but some issues should be improved. Some of the comments that would improve the overall quality of the study are:

a. Authors must pay attention to the technical terms acronyms they used in the text.

b. English language needs to be revised.

c. Conclusion Section: This paragraph required a general revision to eliminate redundant sentences and to add some "take-home message".

Author Response

DEAR REVIEWER, THANKS FOR PROVIDING YOUR CONSIDERATIONS.

MUCH APPRECIATED, AS YOU HELPED ME TO MAKE MY MANUSCRIPT BETTER

I WILL ANSWER IN CAPITAL LETTER

MODIFIED TEXT IS HIGHLIGHTED IN YELLOW

The authors aimed, in a narrative review to focus about what is known in the antibiotics posology and clearance in critically ill with renal failure in RRT. The relevance of such a question depends on the RRT device filter clearance and on antibiotics  steric hindrance.

The study covers some issues that have been overlooked in other similar topics. The structure of the manuscript appears adequate and well divided in the sections. Moreover, the study is easy to follow, but some issues should be improved. Some of the comments that would improve the overall quality of the study are:

a. Authors must pay attention to the technical terms acronyms they used in the text.

I HAVE PROVIDED

b. English language needs to be revised.

I HAVE REVISED

c. Conclusion Section: This paragraph required a general revision to eliminate redundant sentences and to add some "take-home message".

RE-WRITTEN AND REDUCED ALL THE PARTS, HIGHLIGHTING TAKING HOME MESSAGE

Reviewer 3 Report

Interesting article on the use of antibiotics in patients with acute renal failure.

This manuscript gives us an interesting overview of the generalities of antibiotics and the new commercialized antibiotics, generalities of acute renal failure, the correct management of antibiotics, dosing, and route of administration, according to the properties and PK / PD of antibiotics.

The explanation of the acronyms in the abstract is missing

The definitions of the acronyms appear after their first appearance, the initials are not mentioned or do not coincide, for example, Acute Renal Failure (IRA)

There are some small typing errors

need "ofa" CRRT make highly  (line 17)

…but bactericidal antibiotics "t" high doses, which inhibit (line 159)

….fluoroquinolones "a re" Finafloxacin and Zabofloxacin. [7]. (line 210)

Author Response

DEAR REVIEWER, THANKS FOR PROVIDING YOUR CONSIDERATIONS.

MUCH APPRECIATED, AS YOU HELPED ME TO MAKE MY MANUSCRIPT BETTER

I WILL ANSWER IN CAPITAL LETTER

MODIFIED TEXT IS HIGHLIGHTED IN YELLOW

Interesting article on the use of antibiotics in patients with acute renal failure.

This manuscript gives us an interesting overview of the generalities of antibiotics and the new commercialized antibiotics, generalities of acute renal failure, the correct management of antibiotics, dosing, and route of administration, according to the properties and PK / PD of antibiotics.

The explanation of the acronyms in the abstract is missing

CORRECTIONS PROVIDED

The definitions of the acronyms appear after their first appearance, the initials are not mentioned or do not coincide, for example, Acute Renal Failure (IRA)

CORRECTIONS PROVIDED

There are some small typing errors

 CORRECTIONS PROVIDED

 …need "ofa" CRRT make highly  (line 17)

I HAVE CORRECTED

but bactericidal antibiotics "t" high doses, which inhibit (line 159)

CORRECTIONS PROVIDED

.fluoroquinolones "a re" Finafloxacin and Zabofloxacin. [7]. (line 210)

CORRECTIONS PROVIDED

Round 2

Reviewer 1 Report

Please check the below points:

1. Reference numbers 31, 42, 57, and 61 are not cited in the text.

2. Line number 431 - 3.4.3. Basic principles of RRT [510] - 510 is this a reference number? 

3. Please make sure that all the subtitles are arranged as per the author guidelines of the journal